# Ecological Speciation without Morphological Differentiation? A New Cryptic Species of *Diodontus* Curtis (Hymenoptera, Pemphredonidae) from the Centre of Europe [note 1]

**DOI:** 10.3390/insects15020086

**Published:** 2024-01-26

**Authors:** Eduardas Budrys, Svetlana Orlovskytė, Anna Budrienė

**Affiliations:** Institute of Ecology, Nature Research Centre, Akademijos 2, 08412 Vilnius, Lithuania; svetlana.orlovskyte@gamtc.lt (S.O.); ana.budriene@gamtc.lt (A.B.)

**Keywords:** ecological speciation, endosymbionts, differentiation by habitat, multi-gene phylogeny, species delimitation

## Abstract

**Simple Summary:**

The application of molecular methods in the studies of biological diversity revealed that there are many more species than we can recognise by their appearance. Species pairs or groups, which are difficult to recognise and are known as cryptic species, may differ in their ecological traits, place in trophic networks, and their functional role in ecosystems. It is important to separate cryptic species, particularly if we apply species composition or other diversity metrics for an assessment or monitoring of the ecosystem state or human pressures and impacts. In our study, we revealed a new cryptic species of aphid-hunting wasp that is virtually indistinguishable from a well-known one by morphology. These two species can be easily segregated using molecular characters, and presumably they differ also by nesting habitat.

**Abstract:**

Upon exploring the mitotype diversity of the aphid-hunting wasp, *Diodontus tristis*, we revealed specimens with highly divergent mitotypes from two localities in Lithuania and nesting in clayey substrate, while the specimens with typical mitotypes were found nesting in sandy sites. The comparison of inter- and intra-specific distances and application of delimitation algorithms supported the species status of the clay-nesting populations. Using a set of DNA markers that included complete or partial sequences of six mitochondrial genes, three markers of ribosomal operon, two homeobox genes, and four other nuclear genes, we clarified the phylogenetic relationships of the new cryptic species. The endosymbiotic bacteria infestation was checked, considering the option that the divergent populations may represent clades isolated by *Wolbachia* infection; however, it did not demonstrate any specificity. We found only subtle morphological differences in the new clay-nesting species, *D. argillicola* sp. nov.; the discriminant analysis of morphometric measurements did not reliably segregate it as well. Thus, we provide the molecular characters of the cryptic species, which allow confident identification, its phylogenetic position within the genus, and an updated identification key for the *D. tristis* species group.

## 1. Introduction

Cryptic species, which are closely similar or identical in morphology, but different in molecular, behavioural, ecological, and other characteristics, represent high levels of undiscovered biodiversity [1,2,3]. Hence, the rigorous assessment of cryptic biodiversity is crucial for both a deeper understanding of the evolutionary processes of speciation and recognising species limits in order to inform their research or conservation actions [4].

Accelerating investigation into cryptic species’ diversity in the past decades has revealed that modern molecular-based delimitation methods provide valuable complements to morphological taxonomy [2,5,6,7]. Especially useful is an integrative analytical approach, incorporating a range of species’ traits including the behavioural, ecological, and life history aspects [3,8,9,10]. Recently, the integration of morphometrics and molecular methods has been used for the differentiation of cryptic species among the multiple taxa of wasps also [11,12,13,14,15,16,17].

*Diodontus* Curtis, 1834 (Pemphredonidae; family status after [18]), is a genus of predatory solitary apoid wasps with 78 described species worldwide, two thirds of which inhabit the Palearctic region [19]. Females of this genus dig unicellular or multicellular tunnel nests primarily in clay or sand, provisioning them with Aphididae (Hemiptera) as larval food [20,21]. Apoid wasps, including Pemphredonidae, fulfil ecosystem services of biological control (as important natural enemies of herbivorous insects) and to a lesser extent, of pollination [20,22,23,24,25]. Most of them depend on specific natural or early- or late-successional habitats, and thus they may serve as ecological indicators of anthropogenic impact [26].

At present, a total of 12 accepted species of the genus *Diodontus* are known in continental Europe [19,27,28]. However, a preliminary analysis of barcoding sequences available from the Barcode of Life Data System (BOLD) has revealed that the Mediterranean complex of species related to *D. insidiosus* Spooner, 1938, is represented by multiple divergent and geographically restricted mitotypes forming separate Barcode Index Numbers (BINs), which may correspond to undescribed cryptic species [28]. Upon exploring the mitotype diversity of *D. tristis* (Vander Linden, 1829) in Lithuania, we also revealed specimens with highly divergent mitotypes, which formed a separate cluster in phylogenetic analyses.

Among various potential drivers that have been associated with mitotype divergence in arthropods, ecological factors and the effects of maternally inherited endosymbiotic bacteria have been suggested [29]. Due to their general biological similarity, cryptic species may compete for limited ecosystem resources and exclude each other [14,30,31,32], particularly in cases of recent phylogenetic divergence [4]. Alternatively, they may coexist as a result of either sharing of unlimited common resources [12] or due to niche partitioning and ecological divergence along with behavioural or physiological reproductive isolation [33,34], particularly if their similarity is the result of morphological convergence or stasis [4].

Unlike the cryptic divergence generated by ecological factors, the contribution of endosymbiotic bacteria to the evolutionary diversification of the host species still raises intriguing questions and debates [35,36]. Reproductive manipulators, such as *Wolbachia* Hertig, 1936 (Alphaproteobacteria), *Arsenophonus* Gherna et al., 1991 (Gammaproteobacteria), and *Spiroplasma* Saglio et al., 1973 (Mollicutes), in order to enhance their own vertical transmission, may reduce the reproductive output of uninfected females relative to the infected ones due to induced cytoplasmic incompatibility, or cause female-biased sex ratios by killing male offspring [37]. Such reproductive manipulations may reduce the gene flow between populations, leading to increasing reproductive isolation and even speciation triggered by bacterial endosymbionts [38,39,40,41]. The prevalence and effects of endosymbionts on Hymenoptera, including solitary wasps, remains relatively little explored [42].

Therefore, the aim of this study is to test the hypotheses that the cluster of divergent *Diodontus tristis* mitotypes represents a reproductively isolated population infected by a specific strain of endosymbionts or that it is a new undescribed cryptic species.

## 2. Materials and Methods

### 2.1. Specimens

The specimens were collected using an entomological net at their nesting site on a clay wall of a daubed building or while feeding on the honeydew of *Panaphis juglandis* (Goeze, 1778) on *Juglans regia* leaves. They were fixed in 96% ethanol, and a part of them was pinned after DNA extraction. The specimens and the samples of DNA are deposited in the collection of the Nature Research Centre, Vilnius (NRCV).

### 2.2. Morphometric Measurements

The measurements were performed using an ocular micrometer in a Nikon SMZ1000 binocular microscope (www.nikon.com, accessed on 10 September 2020). Images were obtained by means of a digital camera (Nikon DS-Fi2) connected to the microscope; they were stacked using the software CombineZP, version 1.0 (Alan Hadley; https://alan-hadley.software.informer.com, accessed on 15 January 2023).

We used the morphometric measurements described in earlier studies [28,43]: width of head (WH) was defined as the maximum visible width in the dorsal or frontal view; length of face (LF)—the distance between the fore margin of fore ocellus and the midpoint of the lower margin of clypeus when viewed at a right angle; lower interocular distance (LID)—the shortest distance between the inner margins of the eyes approximately at the level of the antennal sockets in the frontal view; upper interocular distance (UID)—the shortest distance between the inner margins of the eyes approximately at the level of the fore ocellus in the frontal view; length of vertex (LV)—the shortest distance between the hind margin of the fore ocellus and the midpoint of the occipital carina when viewed at a right angle; post-ocellar distance (POD)—the shortest visible distance between the inner margins of the hind ocelli when measured at a right angle; oculo-ocellar distance (OOD)—the shortest visible distance between the outer margin of the hind ocellus and the margin of eye when measured at a right angle; intermandibular distance (IMD)—the distance between the outer margins of the genal swellings just above the fore mandibular condyles in the anterodorsal view; length of clypeus (LCL)—the distance between the midpoint of the frontoclypeal suture and the midpoint of the lower margin of the clypeus when viewed at a right angle; width of clypeal apex (WCA)—the distance between the tips of the lateral teeth on the lower margin of the clypeus; length of scape (LSC)—the maximum visible length of the scape; length of flagellomeres 1–3 (3FL)—maximum visible combined length of the first three flagellomeres without the pedicel; length of flagellomere 6 (L6F)—the maximum visible length; width of flagellomere 6 (W6F)—the maximum visible width without the layer of trichoid sensilla; width of pronotum (PRN)—the distance between the tips of the pronotal lobes in the dorsal view; width of pronotal collar (COL)—the distance between the lateral edges of the transverse carina of the pronotal collar in the anterodorsal view.

The Kolmogorov–Smirnov test for normality of measurement distribution, analysis of variance, and canonical discriminant analysis of the measurements were performed using Statistica, version 8 (Statsoft, Tulsa, OK, USA).

### 2.3. Selection of Molecular Characters

We used the following DNA marker set, a major part of which was applied in our earlier study [28]: two mitochondrial DNA markers—namely a continuous 2650–2664 base pair (bp) long sequence that included complete cytochrome c oxidase subunit 1 (*CO1*) and subunit 2 (*CO2*) and ATP synthase subunit 8 (*ATP8*) with transfer RNA genes (partial *tRNA-Trp*, complete *tRNA-Leu*, *tRNA-Asp*, and *tRNA-Lys*), and non-coding interspaces between them and a continuous 2022–2049 bp long sequence that included complete NADH dehydrogenase subunit 6 (*ND6*), cytochrome b (*CytB*) genes, and partial (3’ end) reverse NADH dehydrogenase subunit 1 (*ND1*) gene with the transfer RNA gene *tRNA-Ser* and non-coding interspaces between them. We also analysed the nuclear DNA markers, including the ribosomal RNA operon, namely partial small subunit (*18S*) and large subunit (*28S*) sequences and internal transcribed spacer 1 (*ITS1*) with a combined length 1973–2020 bp as well as selected protein-coding genes (Table 1). Among the latter, we studied the homeobox genes that control the early development of morphological character-rich anterior and posterior body segments, namely *proboscipedia* (*PB*) and *abdominal B* (*AbdB*). The other markers were chosen from a recently published set of sequences of 3256 protein-coding genes, obtained from the whole-body transcriptomes of 167 species of Hymenoptera [44]. After a preliminary assessment of the overall mean distance among the sequences as a proxy for the evolutionary rate, using MEGA version 7 [45], we selected exon-primed intron-crossing (EPIC) markers in two highly conserved genes, namely ubiquitin-conjugating enzyme E2 G1 (*Ube2g1*) and arginine kinase (*ArgK*), and two highly variable genes, namely mitochondrial ribonuclease P protein 3 (*MRPP3*) and pentatricopeptide repeat domain 2 (*PTCD2*). The exon-intron structure of these genes was reconstructed comparing them with the structure of orthologous genes of *Apis mellifera* Linnaeus, 1758 (Hymenoptera: Apidae), available at GenBank (www.ncbi.nlm.nih.gov/genbank, accessed on 25 September 2023) [46].

We attempted to study the internal transcribed spacer 2 (*ITS2*), as well, however, it could not be sequenced successfully even in homozygous male samples of *Diodontus argillicola* sp. nov. and the related *D. tristis* due to the presence of several tandem repeat regions, the length of which presumably varied among the multiple copies of the spacer sequence inside the same organism.

In the phylogenetic analysis based on the full set of used DNA markers, we included other *Diodontus* species with available ethanol-fixed specimens, namely *D. tristis*, *D. medius* Dahlbom, 1844, *D. minutus* (Fabricius, 1793), *D. polytylus* Budrys, 2019, and *D. guichardi* Budrys, 2019. The phylogenetic relationships of *D. argillicola* sp. nov. with the remaining *Diodontus* were clarified using the *CO1–5′* barcode sequences publicly available from the Barcode of Life Data System [47,48].

**Table 1 insects-15-00086-t001:** DNA markers used as molecular characters for *Diodontus* species identification, phylogeny reconstruction, and assessment of endosymbiotic bacteria infestation.

Abbreviation	Description	Length (bp)	Source
	Mitochondrial protein-coding genes:		
*CO1–5’*	Cytochrome c oxidase, subunit 1, from 5′ end to 715 bp	715	[28]
*CO1–3’*	Cytochrome c oxidase, subunit 1, from 710 bp to the 3′ end	825–828	[28]
*CO2*	Cytochrome c oxidase, subunit 2, complete sequence	678	[28]
*ATP8*	ATP synthase, subunit 8, complete sequence	149	new
*ND6*	NADH dehydrogenase, subunit 6, complete sequence	502–535	[28]
*CytB–5’*	Cytochrome b, from 105 to 859 bp	754	[28]
*ND1*	Cytochrome b, 3′ end from 885 bp, and NADH dehydrogenase, subunit 1, 335 bp long 3′ end (considering reverse translation comparing to CytB)	665	[28]
	Nuclear ribosomal DNA operon:		
*18S*	18S rDNA partial sequence, including V2–V4 variable regions	797–798	[28]
*28S*	28S rDNA partial sequence, including D2–D3 variable regions	683–685	[28]
*ITS1*	Internal transcribed spacer 1, partial sequence	491–538	[28]
	Nuclear protein-coding genes:		
*PB*	*Proboscipedia* gene, partial sequence	500–513	[28]
*AbdB*	*Abdominal B* gene, partial sequence	648	[28]
*ArgK*	Arginine kinase gene, partial sequence	599–605	[28]
*MRPP3*	Mitochondrial ribonuclease P protein 3 gene, partial sequence	678–691	[28]
*Ube2g1*	Ubiquitin-conjugating enzyme E2 G1 gene, partial sequence	597–614	[28]
*PTCD2*	Pentatricopeptide repeat domain 2 gene, partial sequence	664–668	[28]
*16S (Bact.)*	16S rDNA of endosymbiotic bacteria, partial sequence	745–785	[28]
*wsp*	*Wolbachia* outer surface protein, partial sequence	577–589	[49]

### 2.4. DNA Extraction, Polymerase Chain Reaction, and Sequencing

The total genomic DNA was extracted from a piece of thoracic muscle tissue of air-dried or preserved in 96% ethanol specimens, according to the protocol of the GeneJet Genomic DNA Purification Kit (Thermo Fisher Scientific Baltics, Vilnius, Lithuania) or using the method described in [50]. The Polymerase Chain Reaction (PCR) mixture included 12.5 μL of 2× DreamTaq PCR Master Mix (Thermo Fisher Scientific Baltics, Vilnius, Lithuania), 2.5 μL of 10 pmol/μL each primer, 1 μL of genomic DNA, and deionised water up to a total volume of 25 μL. We used T3– and T7–tailed forward and reverse PGR primers, which were constructed using an online BiSearch Primer Design and Search Tool [51] and synthesised at Macrogen (Seoul, South Korea). The primers and PCR conditions are presented in [28,49], except for the new ATP8 marker. The latter was amplified using new primers, T3DioCO2-f2 (5′-attaaccctcactaaagttattggaaaycaatgataytgaag-3′) and T7PemATP6-r1 (5′-aatacgactcactatagatggatcaaaaatwgaaaataaattag-3′), under the following PCR conditions: an initial denaturation at 95 °C for 5 min; 50 cycles of denaturation at 94 °C for 1 min; annealing at 43 °C for 1 min; extension at 68 °C for 1.5 min with final extension at 68 °C for 5 min.

The PCR products were electrophoresised on 1.5% agarose gel (Thermo Fisher Scientific Baltics, Vilnius, Lithuania) with 10,000× GelRed (Biotium, Fremont, CA, USA) and purified following the protocol of Exonuclease I and FastAP Thermosensitive Alkaline Phosphatase (Thermo Fisher Scientific Baltics, Vilnius, Lithuania). The sequencing was performed at the Macrogen company using the BigDye Terminator version 3.1 Cycle Sequencing Kit (Applied Biosystems, Foster City, CA, USA). For sequencing, 17–mer universal primers T3 (5′-attaaccctcactaaag-3′) and T7 (5′-aatacgactcactatag-3′) were applied. The obtained new DNA sequences were deposited in GenBank (Table 2).

### 2.5. Phylogeny Reconstruction and Species Delimitation

The nucleotide sequences were aligned, using the BioEdit 7.2.5 software [52], either with the application of the internal two-sequence alignment or the ClustalW multiple alignment algorithm, with additional manual control. When needed, the sequences were converted from Fasta to Nexus format using Mesquite 3.51 [53]. The inter- and intra-specific Kimura 2-parameter (K2P) distances were estimated, and the most likely nucleotide substitution model was selected from the list of 24 available models, using MEGA version 11 [54] and applying the maximum likelihood method. In all cases, the substitution patterns were best described by the general time reversible model with gamma distribution evolutionary rates, applying four discrete categories, and a fraction of invariable sites (GTR+G+I). We applied this model for the reconstruction of phylogenetic relationships using the maximum likelihood algorithm, implemented in MEGA 11, and the method of Bayesian inference, implemented in MrBayes, version 3.2.7 [55]. We treated each gene and where present, each exon and intron of the gene as a separate partition. The nucmodel parameter was set to “codon” for the partitions of exons of mitochondrial and nuclear protein-coding genes; the nucmodel parameter was set to “4by4” for the partitions of tRNA, rRNA, internal transcribed spacer, and introns. For the reconstruction of phylogenetic relationships, Markov Chain Monte Carlo (MCMC) analyses were run for 5,000,000 generations, with tree sampling every 10,000 generations. The convergence between runs was considered sufficient after the standard deviation of the split frequencies reached a value of 0.01. In reconstructing the phylogeny with the application of the maximum likelihood algorithm, we used 10,000 bootstrap replicates.

For species delimitation, we applied three algorithms: the single threshold General Mixed Yule Coalescent model (GMYC) [56], implemented as *splits* package in R environment, the Bayesian Poisson Tree Processes method (bPTP) web server (species.h-its.org/ptp, accessed on 2 June 2023) [17], and the Assemble Species by Automatic Partitioning (ASAP) web server (bioinfo.mnhn.fr/abi/public/asap/asapweb.html, accessed on 2 June 2023) [57]. The ultrametric dichotomous trees for the GMYC analysis were obtained using the RelTime-ML algorithm, implemented in MEGA 11. The results of species delimitation were compared with the Barcode Index Number (BIN) database clustering available from the Barcode of Life Data System [47,48].

## 3. Results

### 3.1. Identification

The specimens with divergent mitotypes were morphologically hardly separable from the typical *Diodontus tristis*. The most useful differences are included in the identification key (Section 3.4 and Section 3.5). The statistically significant morphometric differences are presented in the diagnosis (Section 3.4). The discriminant analysis of morphometric measurements did not allow a reliable segregation of either sex of the genetically divergent specimens from *D. tristis* (Figure 1).

However, these specimens could be reliably separated by multiple characters of mitochondrial and nuclear DNA. Among them, the *CO1–5’* barcode and its part—the 132 bp long “minibarcode” (Figure 2) that is easier to obtain from dry collection specimens than the full barcode—as well as the sequences of other studied mitochondrial genes (*CO2*, *ATP8*, *ND6*, *CytB*) and the nuclear markers, including the intron of a conserved *proboscipedia* (*PB*) homeobox gene that differs from that of *D. tristis*, in addition to several single nucleotide differences via a typical of the new species, 7 bp long deletion at position 130 (Figure 3).

### 3.2. Phylogenetic Relationships and Species Delimitation

The phylogenetic analysis of all barcoding *CO1–5′* mitotypes of *Diodontus*, obtained in this study or available from the BOLD BIN database, using the Bayesian inference and the maximum likelihood algorithms, demonstrated that the divergent mitotype cluster is a sister group of the cluster of the remaining mitotypes of *D. tristis*. The application of three species delimitation algorithms, GMYC, bPTP and ASAP, to the available barcoding sequence set of genus *Diodontus* confirmed the distinctiveness of the cluster and its status as a separate cryptic species, *D. argillicola* sp. nov. (Figure 4).

The mean inter-specific distance between the two species, estimated using the 658 bp long barcoding sequence and K2P substitution model, was 0.116, while the intra-specific distances among the mitotypes were 0.006 for *D. argillicola* sp. nov. and 0.002 for *D. tristis*, thus implying the presence of a barcoding gap. The mitotype FBACA563-10 from the BOLD BIN database (Figure 4), most probably representing one more cryptic species of the *D. tristis* complex, was not included in this analysis.

The distinctiveness of the new species was validated via an application of the delimitation algorithms to the mitotypes of *D. argillicola* sp. nov. and *D. tristis* based on a 4719 bp long mtDNA sequence of complete genes *CO1*, *CO2*, *ATP8*, *ND6*, *CytB*, and partial *ND1* (Figure 5a). The mean inter-specific distance between the two species was 0.101, while the mean intra-specific distance among the studied mitotypes of *D. argillicola* sp. nov. was 0.006, and the distance among the available mitotypes of *D. tristis* was 0.001. The distinctiveness was confirmed by differences in all the studied nuclear markers, including a unique deletion in the *proboscipedia* (*PB*) gene’s lesser intron at the position 130 (Figure 3).

The summarizing reconstruction of phylogenetic relationships using the Bayesian inference and the maximum likelihood algorithms and the 10,521 bp long combined set of mitochondrial and nuclear DNA markers confirmed the relatedness of *D. argillicola* sp. nov. and *D. tristis*; *D. medius* turned out to be a more phylogenetically distant member of the *D. tristis* species group (Figure 5b).

**Figure 2 insects-15-00086-f002:**
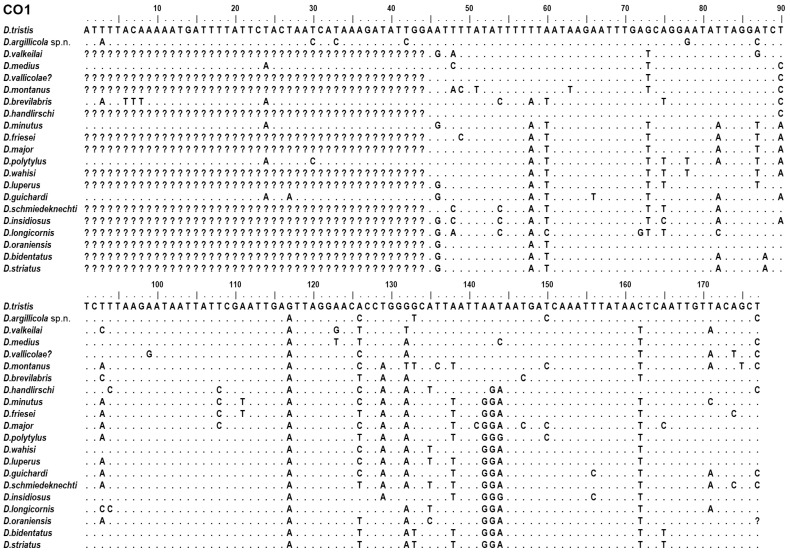
Comparison of the 5′ end of mitochondrial cytochrome c oxidase, subunit 1, and the “minibarcode” sequence of *Diodontus argillicola* sp. nov. with those of *D. tristis* and other species of the genus, where this molecular character is available. Dots—identical nucleotides; question marks—unknown sequences. *D. vallicolae?* represents the sequence of an unidentified specimen with BOLD processID AMCAO361-20 (see also Figure 4) that likely belongs to *D. vallicolae* Rohwer, 1909.

**Figure 3 insects-15-00086-f003:**
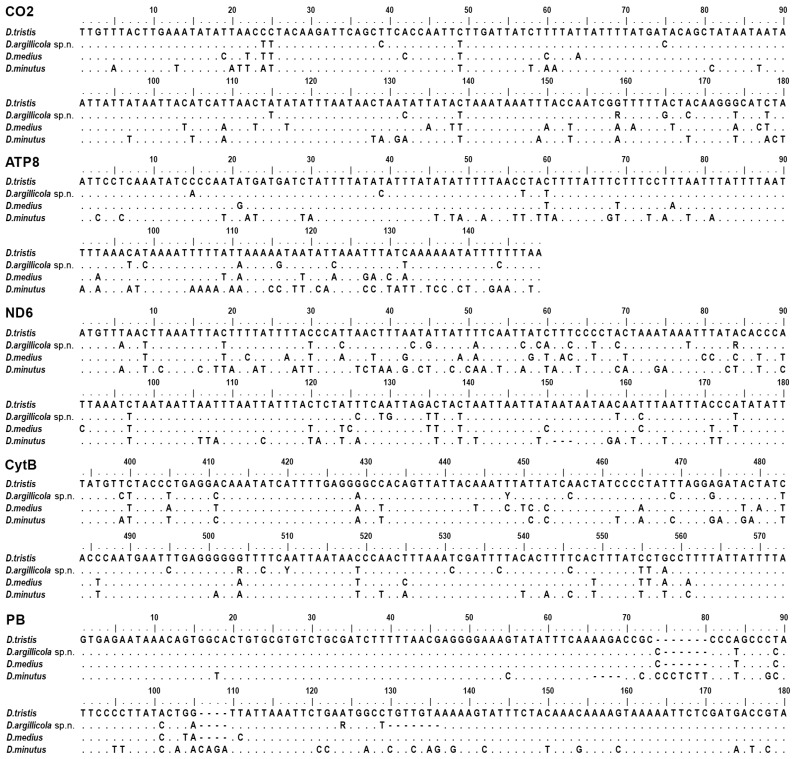
Comparison of selected molecular characters (180 bp long consensus sequences) of *Diodontus argillicola* sp. nov. with those of *D. tristis*, *D. medius*, and *D. minutus*: CO2, 5′ end of mitochondrial cytochrome c oxidase, subunit 2; ATP8, complete sequence of mitochondrial ATP synthase, subunit 8; ND6, 5′ end of mitochondrial NADH dehydrogenase, subunit 6; CytB, the fragment of mitochondrial cytochrome b, commonly used as the CytB barcode; PB, almost complete sequence of the lesser intron of *proboscipedia* homeobox gene. Dots—identical nucleotides; dashes—deletions.

**Figure 4 insects-15-00086-f004:**
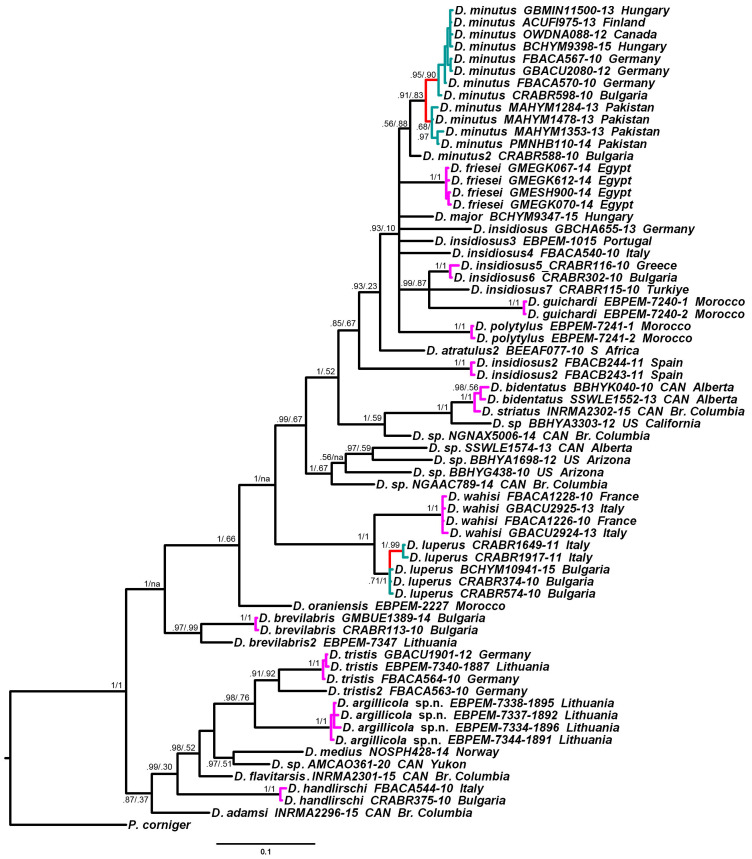
Phylogenetic relationships of *Diodontus*, reconstructed using GTR+G+I evolution model, applied to the barcoding *CO1–5′* sequence of mtDNA. Tree topology reconstructed using Bayesian inference; statistics of branches: Bayesian posterior probability (5,000,000 generations)/maximum likelihood probability (10,000 bootstrap replicates). Mitotype labels include the specimen process ID codes of the Barcode of Life Data System or (the labels including EBPEM) the unpublished barcode sequences from the database of the authors and the collection country. The mitotype clusters supported as species by delimitation algorithms: magenta—GMYC, bPTP, and ASAP; red—GMYC; teal—bPTP and ASAP. *Passaloecus corniger* Shuckard, 1837, was used as an outgroup. The scale: probability of expected nucleotide changes per site.

**Figure 5 insects-15-00086-f005:**
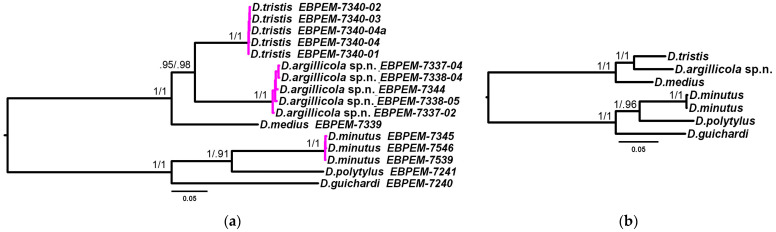
Phylogenetic relationships among *Diodontus argillicola* sp. nov. and closely related species *D. tristis* and *D. medius*. Tree topology reconstructed using Bayesian inference; branch statistics: Bayesian posterior probability (5,000,000 generations)/maximum likelihood probability (10,000 bootstrap replicates). *D. minutus* species group (*D. minutus*, *D. polytylus*, and *D. guichardi*) was used as an outgroup. (**a**) reconstruction using the markers of mitochondrial DNA, 4719 bp long sequence including complete *CO1*, *CO2*, *ATP8*, *ND6*, *CytB*, and partial *ND1* genes; the magenta haplotype clusters were supported as species by delimitation algorithms GMYC, bPTP, and ASAP. (**b**) reconstruction using full 10,521 bp long set of studied mitochondrial and nuclear markers (see Table 1). The scale: probability of expected nucleotide changes per site.

### 3.3. Endosymbiotic Bacteria Infestation

Using the *16S* marker, we explored the infestation of muscle tissues of *Diodontus* wasps by intra-cellular symbiotic bacteria. Nearly all studied specimens were infected by *Wolbachia*. One *D. tristis* specimen was infected by *Spiroplasma*, the *16S* partial sequence of which was very similar (99.7% similarity) to the sequence with GenBank ID LC388759.1 that was isolated from ticks. Three *D. minutus* specimens were infected by *Arsenophonus*; the *16S* partial sequence of which was most similar (99.9%) to the sequence with GenBank ID DQ115536.1, isolated from a hippoboscid fly.

Most *D. tristis* and *D. minutus* individuals had an infection of typical *Wolbachia pipientis* Hertig, 1936, with a partial sequence of outer surface protein (*wsp*) identical to that of, for instance, *Apanteles chilonis* Munakata, 1912 isolate (GenBank sequence ID KC161914.1). The 2 studied *D. tristis* and all 10 studied *D. argillicola* sp. nov. had a mixed *Wolbachia* infection, represented by two *wsp* gene strains, which were presumably identical to those with GenBank sequence IDs KC161914.1 and JN639221.1; the latter sequence was isolated, among others, from *Andrena wilkella* (Kirby, 1802).

We did not find any endosymbiotic bacteria strains specific only to *D. argillicola* sp. nov. or *D. tristis*. Therefore, we rejected the hypothesis that the new species represents a clade of *D. tristis*, which is reproductively isolated due to specific endosymbiont infestation.

### 3.4. Diodontus argillicola Budrys, Orlovskytė & Budrienė, New Species

Zoobank link: urn:lsid:zoobank.org:act:856E17E8-C310-4BF2-B459-D6DF115BD1ED

Figure 6, Figure 7 and Figure 8a,b,e.

Type material. Holotype: ♀, Lithuania, Ūdininkai, clay wall of daubed building (54°13’ N 23°24’ E), 13.vi.2015, E. Budrys leg. (deposited in NRCV, voucher No EBPEM-7338-6). Paratypes: Lithuania: same data as holotype, 6♀15♂, E. Budrys, S. Orlovskytė leg. (NRCV, voucher Nos EBPEM-7338-01–21); same locality and habitat as holotype, 5♀3♂, 04.vii.2013 (NRCV, voucher Nos EBPEM-7337-01–08); 1♂, 11.vii.2019 (NRCV, voucher No EBPEM-7631); Bilšiai (55°08′ N 25°16′ E), honeydew on *Juglans regia* leaves, 1♂, 27.viii.2019, all E. Budrys leg. (NRCV, voucher No EBPEM-7344).

Consensus barcode: aattttatattttttaataagaatttgagcagggatattaggctcttctttaagaataattattcgaattgaattaggaacmcctgggtcattaattaataatgaccaaatttataactcaattgttacagcccatgcttttattataattttttttatagtaataccttttataattggagggtttggaaattgattaatccccataataattggrgcccctgatatggctttcccccgaataaataatataagattttgrttaatccccccctctctttttattttaattttaagaaacattttaaataatggggtagggacaggttgaacagtttacccccccttatccgctaatattagycataatggatcttcagtagatctcgctattttttctttacatattgcaggggtrtcatcaattataggrgctattaattttattgttacaattataaacataaaaaataaattcttaaattttgatcaaatacccttatttgtytgatcaatcttratcacaaccatccttttactcctatccttrccagtccttgcaggagcaattacaatattattaactgaccgaaatttaaatacaactttttttgacccatctggagggggggayccaattttatatcaacatttattt

Distribution. Currently known from two localities in Lithuania.

Etymology. Species name is derived from the substrate where the species was observed nesting, clay (Ancient Greek αργιλλος, “clay”) and the Latin word *cola*, which means “inhabitant” or “dweller”. A noun is used in apposition to the generic name.

Diagnosis. Both sexes of *D. argillicola* sp. nov. are very similar to *D. tristis* in body sculpture, punctation, and colouration. The morphometric differences are very weak (Figure 1; Table 3). A reliable tool for segregation of the two species is molecular barcoding (Figure 2 and Figure 3). The two species seem to also differ behaviourally regarding their choice of nesting substrate: most of the specimens of the new species were collected from a colony nesting in a clay wall, while all barcoded specimens from the colonies nesting in sand or gravel substrate happened to be *D. tristis*.

**Figure 6 insects-15-00086-f006:**
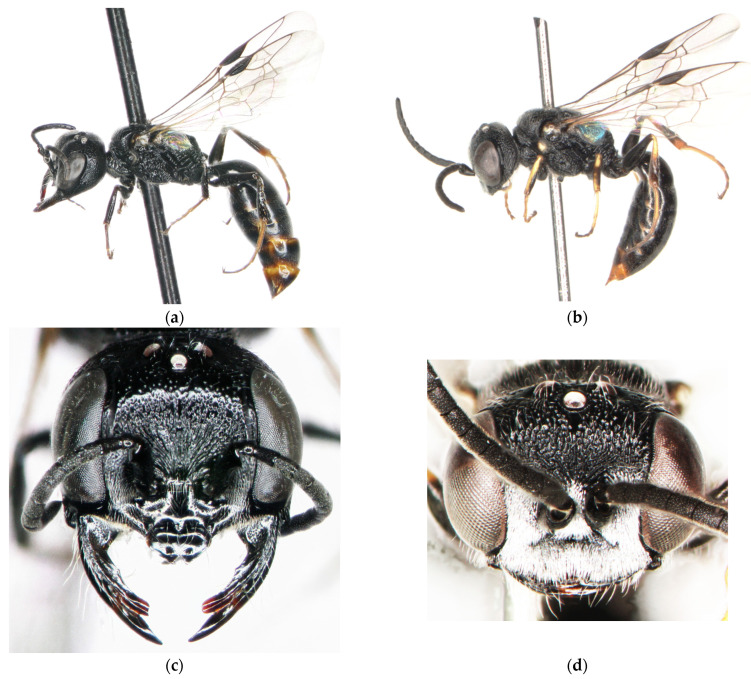
*Diodontus argillicola* sp. nov.: general view, (**a**) holotype female; (**b**) male; head, frontal view, (**c**) female; (**d**) male; thorax, lateral view, (**e**) female; (**f**) male.

**Figure 7 insects-15-00086-f007:**
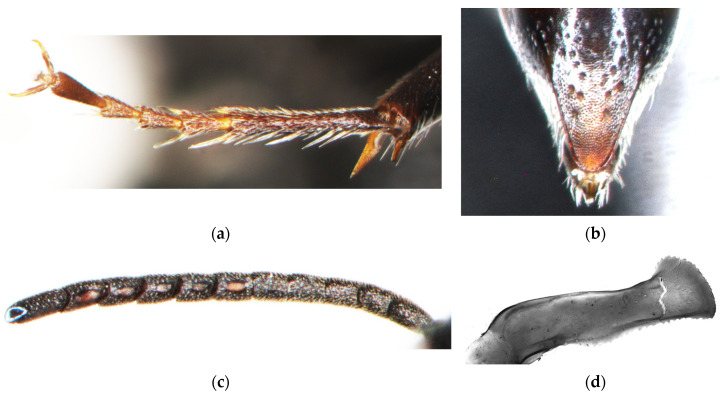
*Diodontus argillicola* sp. nov., female: (**a**) foretarsus; (**b**) pygidial plate; male: (**c**) flagellum; (**d**) penial valve.

**Figure 8 insects-15-00086-f008:**
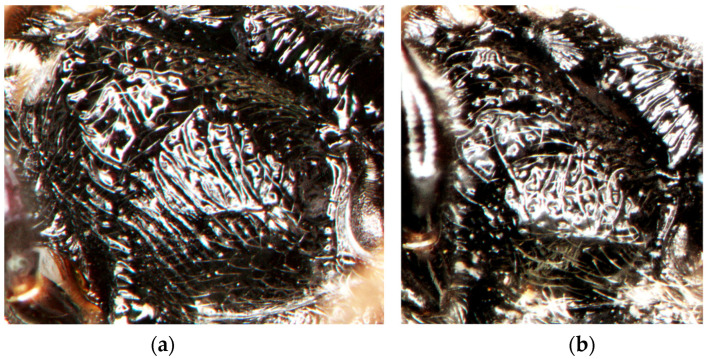
Mesopleuron: *Diodontus argillicola* sp. nov., (**a**) female; (**b**) male; *D. tristis*, (**c**) female; (**d**) male. Female, upper frons: (**e**) *D. argillicola* sp. nov. (two specimens); (**f**) *D. tristis* (two specimens).

*Diodontus argillicola* sp. nov. differs from *D. tristis* by a generally slightly smoother sculpture and sparser punctation, although the variability of these characters overlaps in the two species. In *D. argillicola* sp. nov., the upper frons between the fore ocellus and the frontal gland is slightly smoother, shinier, and more sparsely punctate (Figure 8e) than in *D. tristis* (Figure 8f): the interspaces among punctures are mostly wider than or equal to but rarely narrower than punctures, while in *D. tristis*, the interspaces here are mostly narrower than the width of punctures, sometimes equal to them, and rarely wider. In *D. argillicola* sp. nov., the head of the female is slightly less transverse and more rounded than in *D. tristis*; the dorsal surface of the frons between the fore ocellus and anterior vertical surface is wider than the fore ocellus (Figure 8e), while in *D. tristis*, it is approximately equal to the fore ocellus (Figure 8f). The subscrobal area of the mesopleuron is more regularly obliquely to nearly transversely rugose; this rugosity is less regular below the posterior end of the scrobus (Figure 8a,b), however it is not so irregularly areolate as in *D. tristis* (Figure 8c,d).

Morphometric differences of *Diodontus argillicola* sp. nov. from *D. tristis*. In the female, head of *D. argillicola* sp. nov. is slightly less transverse with WH : LID = 1.564 ± 0.010 (*N* = 9), while it is 1.583 ± 0.003 (*N* = 73) in *D. tristis* (ANOVA: *F* = 4.3, *p* = 0.04). The oculo-ocellar distance in *D. argillicola* sp. nov. female is relatively smaller with POD : OOD = 0.900 ± 0.019 (*N* = 9), while it is 0.850 ± 0.005 (*N* = 73) in *D. tristis* (ANOVA: *F* = 9.8, *p* = 0.002). In the male of *D. argillicola* sp. nov., in contrast, the post-ocellar distance is comparatively smaller, POD : OOD = 0.912 ± 0.032 (*N* = 7), while it is 0.954 ± 0.004 (*N* = 137) in *D. tristis* (ANOVA: *F* = 4.6, *p* = 0.03). The flagellum of *D. argillicola* sp. nov. is slightly shorter: in the female, 3FL : LSC = 0.89 ± 0.01 (*N* = 9), while it is 0.95 ± 0.01 (*N* = 70) in *D. tristis* (ANOVA: *F* = 10.8, *p* = 0.002); in male, L6F : W6F = 1.30 ± 0.04 (*N* = 7), while it is 1.39 ± 0.01 (*N* = 127) in *D. tristis* (ANOVA: *F* = 5.0, *p* = 0.027).

Description.

Female (Figure 6a,c,e, Figure 7a,b and Figure 8a,e). The body length is 6.3–6.9 mm (holotype 6.6 mm). The body is black. In the holotype and a part of paratypes, the mandible is dark yellowish-brown with a dark red tip. The protibia medially, metatibia basally and apically, and tarsi are brown; in other paratypes, these body parts are dark brown. The pterostigma is nearly black with a dark brown strip along the edge of the wing.

The head is weakly transverse (frontal view; Figure 6c), WH : LF = 1.56. The face is moderately wide, LID : LF = 0.89. The inner orbits are not converging ventrad, with LID : UID = 1.02. The ocelli form an obtuse triangle, with POD : OOD = 0.90. The frontal line in front of the fore ocellus is weakly impressed. The area of the interantennal process is nearly flat. Orbital gland is concave, its upper part does not have distinct swelling medially; the mid part is approximately equal or slightly wider than the space between it and the inner orbit. The mandibular condyles are relatively widely separated, IMD : WH = 0.78. The clypeus is relatively long, LCL : LF = 0.26. The clypeal apex is tridentate, comparatively broad, WCA : IMD = 0.33, and the mid tooth is shorter than the lateral teeth. The labrum is subtriangular with a subtriangular apical notch. The scape is moderately long, LSC : LF = 0.46. The flagellum is moderately long, 3FL : LSC = 0.89. The frons is partly rugulose, weakly shiny, and densely punctate to scrobiculate; the interspaces are alutaceous and narrower than the punctures. The oculo-ocellar area of the upper frons is nearly smooth and with scattered punctation; a part of the interspaces is 1–2 times as wide as the punctures (Figure 8e). The subappressed pilosity of the upper frons is sparse and nearly as long as the width of the ocellus. The pilosity of the lower frons between the eye and antennal socket is pale grey, it does not conceal the underlying microsculpture. The upper part of the clypeus has relatively sparse short subappressed pilosity and multiple long setae; the shiny impunctate area above the lower edge covers more than half of the height of the clypeus.

The pronotal collar is moderately wide, COL : PRN = 0.58, it has narrowly rounded sides and weakly convex dorsal carina (frontal view). The lateral surface of the pronotal collar is separated from the dorsal surface by weak carina and rugulose. The anterior part of the scutum is densely punctate; it has interspaces narrower than the punctures. The disc of the scutum is smooth and shiny, with remote, irregularly scattered, and distinct punctation; the interspaces are 2–8 times as wide as the punctures. The scutellum is glabrous and shiny with a few shallow punctures. The episcrobal area of the mesopleuron is coarsely areolate. The subscrobal and ventral parts of the mesopleuron are coarsely, obliquely, and regularly rugose; this rugosity is less regular postero-dorsally below the scrobus (Figure 6e and Figure 8a). The metapostnotum is irregularly rugose. The propodeum is coarsely areolate, with obtuse lateral angles and indistinct, uneven carina separating the lateral and posterior surfaces. The probasitarsal rake is weak, with preapical spines slightly longer than the width of the probasitarsus (Figure 7a).

The gastral terga are smooth and finely punctate: the tergum 1 is strongly shiny, without microsculpture between the punctures; the remaining terga are shiny, with fine alutaceous microsculptures. The pygidial plate is subtriangular, with strong scattered punctures, its surface is weakly shiny with distinct coriaceous to granulose microsculpture (Figure 7b).

Male (Figure 6b,d,f, Figure 7c,d and Figure 8b). The body length is 4.9–5.2 mm. Body is black with ivory spots on the tegula and spiracular lobe. The protibia anteriorly, mesotibia basally and antero-apically, and metatibia basally are pale yellow; in a part of the paratypes, the apical yellow marking of the mesotibia is strip-like elongated and nearly reaching the basal marking. The apical segment of the abdomen is reddish-brown.

The body structure is generally similar to that of the female. The head is moderately transverse, WH : LF = 1.62. The face is moderately narrow, LID : LF = 0.79; the inner orbits are distinctly converging ventrad, LID : UID = 0.85. The frontal line is indistinct,. POD : OOD = 0.91. The clypeal apex is bidentate. The labrum is subtrapezoidal with a weakly emarginate apical margin. The scape is moderately long, LSC : LF = 0.39. The flagellum is comparatively long, 3FL : LSC = 1.39; the flagellomeres are more long than wide, L6F : W6F = 1.30. Flagellomeres 5–10 have angularly protruding apicoventral edges and dark, smooth, flat, ovoid tyloidea; very small tyloid-like tubercles are also discernible on flagellomeres 4 and 11 (Figure 7c). COL : PRN = 0.59. The probasitarsi and mesobasitarsi are nearly straight. The mesobasitarsus is not dilated. Tergum 7 has distinct lateral carinae, forming a subtrapezoidal pygidial plate. The penial valve has a wide denticulate apex (Figure 7d).

The sculpture and pilosity are similar to those in the female, with the following exceptions: punctation is generally denser; interspaces in the oculo-ocellar area are smaller than the punctures; the pilosity of the lower frons and clypeus is very dense, silver, it conceals the underlying microsculpture (Figure 6d). The disc of the scutum has close punctation; the interspaces are shining and 1–2 times as wide as the punctures. The scutellum is closely punctate; the interspaces are as wide as the punctures. The subscrobal part of mesopleuron is less regularly rugose (Figure 8b); its ventral part is nearly smooth and shining. Tergum 6 is without distinct small spines. The pilosity of sterna 3–5 is sparse without specific features.

### 3.5. Key to the Palearctic Species of Diodontus tristis Species Group

The *Diodontus tristis* species group is represented by relatively large and dark species with Palearctic or Nearctic distribution ranges, characterised by the even pilosity of the male’s tergum 6, the lack of two groups of spines on it; such spines, which are clearly longer and thicker than the surrounding setae, are typical of males of the remaining *Diodontus* species.

1.In the male, tergum 6 has uniform pilosity and several longer preapical setae, without two distinct groups of thick spines. In the female, antenna, mandible, tegula, mid and hind tibiae are black or brown, without distinct yellow spots. In doubtful cases (in *D. montanus*, the tibiae are entirely pale brown), the face is very wide: the smallest distance between the eyes at the level of the antennal sockets is equal to the distance between the fore ocellus and the tip of the mid tooth of the clypeus; the distance between the lateral teeth of the clypeal apex is 1.4–2 times the length of the clypeus, and clearly larger than the distance between the lateral tooth and eye; the upper end of the frontal glands is flat or concave (*D. tristis* species group).........................................................................................................................................................................................................................2

-In the male, tergum 6 has two preapical groups of small spines, which are clearly thicker than the surrounding pilosity. In the female, the combination of the listed characters is otherwise. If the mandibles and tibiae are dark, then at least one of the following is true: (a) the face is more rounded: the smallest distance between the eyes at the level of the antennal sockets is less than 0.85 times as wide as the distance between the fore ocellus and the tip of the mid tooth of the clypeus; and/or (b) distance between the lateral teeth of the clypeal apex is not larger than the distance between the lateral tooth and eye; and/or (c) the frontal gland is convex at the upper end.…………………………………………………………………………………………..........................……...…..……...…(other species groups)

2.The frons and posterolateral part of the mesopleuron are smooth, they have sparse weak punctation and coriaceous interspaces. In the female, the pygidial plate is typically widely triangular, covering all the dorsal surface of tergum 6, without a distinct marginal rim; the frontal glands are narrow, with their upper part deeply impressed in relation to the oculo-ocellar area; the tibiae are brown to pale brown. In the male, the pilosity of the lower frons and clypeus is sparse and weakly shining, and it does not hide the surface of cuticle. (Mountains of Central Asia from SE Kazakhstan to Tajikistan) ................................................................................*D. montanus* Kazenas, 1992

-The central part of the frons is strongly and densely punctate. The posterolateral part of the mesopleuron is obliquely or irregularly rugose or areolate. In the female, the pygidial plate is narrower, it is subelliptic with a distinct marginal rim; the upper part of the frontal gland is flat or weakly concave; the mid and hind tibiae are black or dark brown. In the male, the pilosity of the lower frons and clypeus is dense, silvery, and it hides the surface of the cuticle.……………………………………………….......………………………………….....……………...…………........…………………………....3

3.The scutum is finely and very densely punctate; the interspaces are mostly narrower than the punctures. In the female, the frontal gland is twice as wide as the space between the gland and the eye; all tibiae are laterally black and medially brown, without a yellow pattern. In the male, the tegula and spiracular lobe have ivory spots; the interspaces between the punctures on the oculo-ocellar area are coarsely granulose, matt. (Pan-palearctic boreal).....……………………………...............…….....…....…………..*D. medius* Dahlbom, 1844

-The scutum has more or less scattered punctation, particularly in the female; at least medially, the interspaces are wider than the punctures. In the female, the frontal gland is as wide as the space between the gland and eye. In the male, the tegula and spiracular lobe are black (*D. valkeilai* and *D. asiaticus*); if they are with ivory spots (most of *D. tristis* and *D. argillicola*), then the interspaces between the punctures on the oculo-ocellar area are alutaceous and shiny......………………………………….....……………..............................................................................................................................................4

4.The upper frons has dense erect setosity; the setae are longer than the width of the ocellus. In the female, the fore tibia anteriorly has a yellow strip. The face is very wide: the smallest distance between the eyes at the level of the antennal sockets is equal to the distance between the fore ocellus and the tip of the mid tooth of the clypeus. The body length is 8 mm in females and 6.5 mm in males. (Siberia: Chita oblast)......………………………………….....……………..............................................................................................*D. valkeilai* Budrys, 1992

-The upper frons has sparser subappressed setosity; the setae are shorter than the width of the ocellus. In the female, the fore tibia is without a yellow strip; the face is narrower: the smallest distance between the eyes at the level of the antennal sockets is 0.8–0.95 as long as distance between the fore ocellus and the tip of the mid tooth of the clypeus. The body length is typically up to 7 mm in females and up to 6 mm in males.......………………………………….....……………............................................................................................................................................5

5.The oculo-ocellar area has scattered punctation: the interspaces are twice as wide as the punctures in females and 1.5 times as wide as the punctures in males. The posterolateral part of the mesopleuron is more or less regularly obliquely striate. In the female, the upper frons has moderately dense punctation, the interspaces are approximately as wide as the punctures and smooth. In the male, the tegula and spiracular lobe are black, while the hind tibia is black with a yellow base. (Mongolia)........................*D. asiaticus* Tsuneki, 1972

-The oculo-ocellar area has dense punctation: the interspaces are as wide as the punctures or smaller. The posterolateral part of the mesopleuron is more irregularly rugose-areolate. In the female, the upper frons is very densely rugulose-punctate: the interspaces are smaller than the punctures. In the male, the tegula and spiracular lobe are usually (but not always) with ivory spots, and the hind tibia is either black with a yellow base or with more or less extended pale brown colouration antero-medially. (The following two species may be securely separated only using molecular characters)......………………………………….....……………...................................................................................................................................6

6.The posterolateral part of the mesopleuron is more regularly rugose (Figure 8a,b). In the female, the dorsal declivous part of the upper frons between the fore ocellus and anterior vertical surface is slightly longer than the width of the fore ocellus; the area between the fore ocellus and frontal gland is smoother, shinier, and more sparsely punctate; its interspaces are mostly wider than or equal to the punctures (Figure 8e). …………………………......…………………………......………………….......................………......*D. argillicola*, sp. nov.

-The posterolateral part of the mesopleuron is more irregularly areolate (Figure 8c,d). In the female, the dorsal declivous part of the upper frons between the fore ocellus and anterior vertical surface is not longer than the width of the fore ocellus; the area between the fore ocellus and frontal gland is more distinctly undose, less shiny, and more densely punctate; its interspaces are mostly narrower than the punctures (Figure 8f). (Western Palearctic, southern Siberia, Mongolia).........................................*D. tristis* (Vander Linden, 1829)

## 4. Discussion

We have not found any morphological character that would allow a reliable separation of *Diodontus argillicola* sp. nov. from *D. tristis*, thus we may consider them as a cryptic species complex that needs further investigation. However, the two species can be easily segregated by virtually all studied DNA characters, including the *CO1* barcoding sequence (Figure 2 and Figure 3). Thus, our study supports the possibility of a minimalist taxonomic approach, which relies on the application of DNA barcodes to quickly and efficiently identify and describe large numbers of newly discovered species from little studied and species-rich taxa [58]. Such an approach may be particularly useful in the clades where speciation and ecological adaptation and/or sexual selection processes are not related to morphological differentiation—for example, the trophic specialisation of emerging species solely includes the physiological adaptations to food plant chemical defences, or the mating partner is recognised by pheromone or epicuticle chemistry.

On the other hand, we agree that a different barcoding sequence of the mitochondrial *CO1* gene is an insufficient argument for considering the mitotype as a separate species. In addition to the assessment of genetic variability, in combination with species delimitation algorithms, the description should be ideally complemented by information on ecological, behavioural, or life history traits as well as the screening of endosymbionts’ composition [10]. In arthropods, the prevalence of endosymbiotic bacteria has been documented in a range of cryptic species complexes, predominantly in agriculturally significant taxa, challenging the hypothesis that cryptic speciation may have been facilitated by endosymbiont infestations [59,60,61,62].

Our results did not demonstrate a peculiar infection pattern in the studied *D. tristis* and *D. argillicola* sp. nov., thus we currently cannot affirm a possible role of endosymbionts in the *Diodontus* species diversification. However, the discovered variety of infestation (*Arsenophonus*, *Spiroplasma*, and several strains of *Wolbachia*) allow us to imply the phylogenetic incongruence between the *Diodontus* hosts and their endosymbionts. This incongruence may indicate that, in addition to the vertical transmission (from mother to offspring), horizontal transmission events may have taken place, such as a transfer from prey to the predator or a transfer through the non-lethal probing of prey [61]. It has been reported that *Diodontus* females immobilize their aphid prey by malaxation with mandibles instead of stinging [20,63]. Since aphids are well-known hosts of bacterial endosymbionts, including *Wolbachia*, *Arsenophonus*, and *Spiroplasma* [64], the horizontal transfer of endosymbionts in *Diodontus* is notably possible, either during the malaxation of the provisioning female or through the consumption of aphids by the offspring.

Niche differentiation is typically associated with parapatric speciation, where the reduction of gene flow between populations is connected with habitat or mating ground segregation. The preliminary observations that *D. argillicola* sp. nov. nests in clay, while *D. tristis* prefers a sandy or gravel nesting substrate, could indicate the potential contribution of the ecological niche partitioning process in species differentiation. The variability of preferred soil characteristics has been considered as an important factor determining the parallel presence of viable populations of several closely related ground-nesting wasp species [65]. Differences in habitat use have also been demonstrated for sympatric cryptic species of cavity-nesting wasps [12,49], bumble bees [66,67], and mayflies [68].

Our results do not demonstrate any role of endosymbiont infection in the speciation of *D. tristis* and *D. argillicola* sp. nov. However, we cannot fully exclude that their differentiation has occurred through simultaneous niche separation and reproductive isolation due to the effects of endosymbionts, as our study of the endosymbiotic microorganisms with the exploration of just two markers is a preliminary one. There are examples in the ground-nesting bee, *Andrena proxima* (Kirby, 1802) species complex, of parapatric and sympatric distributions, where both *Wolbachia*–induced incompatibilities and ecological differences are thought to maintain species boundaries [69]. To better understand the role of bacterial endosymbionts in the diversification of *Diodontus* species, further studies focusing on the strains of each symbiont and the phylogenies of symbiont infections in relation to host phylogenies are required.

The currently known distribution range of *D. argillicola* sp. nov., restricted to Lithuania, is obviously underestimated, since the specimens of this cryptic species have not been separated from *D. tristis*. The lack of specimens of the new species in the BOLD barcode database implies that the new cryptic species may be missing from western Europe, where the available barcodes come from. This lack also implies that the name *Pemphredon tristis*, the nominal species described by Vander Linden from Belgium, most probably does not belong to the new species described here. The collection of Vander Linden, unfortunately, is lost, and therefore a designation of the neotype of *D. tristis* from the type territory, along with a study of its molecular characters, is needed. The presence of a specimen with deviating mitotype FBACA563-10 in the BOLD database, forming a separate BIN AAN4253, which has been collected in Brandenburg (Germany), along with the typical *D. tristis* (BIN AAN4254), suggests the possibility of a wider cryptic speciation in the *D. tristis* species group that needs further exploration.

In the assessments of anthropogenic pressures on early-successional habitats using the structure of digger wasp communities as an indicator, *D. tristis* has been recorded among common or even dominant species in central Europe [70,71]. Our study demonstrates that the data on the presence and abundance of *D. tristis* may need a reconsideration because the cryptic species of this complex may differ in their reaction to habitat change and human impacts.

## Figures and Tables

**Figure 1 insects-15-00086-f001:**
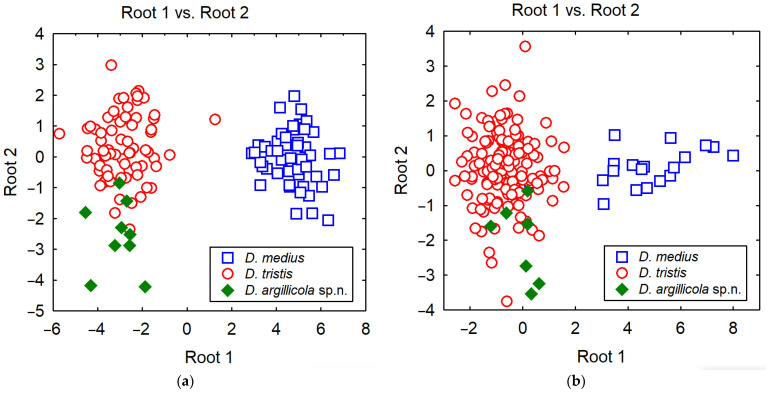
Results of discriminant (canonical) analysis of morphometric measurements (see Section 2.2) of *Diodontus argillicola* sp. nov., *D. tristis* and *D. medius*. Scatterplot of scores of canonical functions: (**a**) females; (**b**) males. Function coefficients are in Table 3.

**Table 2 insects-15-00086-t002:** Accession numbers of DNA sequences of *Diodontus* specimens (vouchers in NRCV) deposited in GenBank (www.ncbi.nlm.nih.gov/genbank, accessed on 10 November 2023).

Species	*D. argillicola* sp. nov.	*D. tristis*	*D. medius*	*D. minutus*
Collection place	54°13′09″ N 23°24′13″ E	55°52′39″ N 23°28′16″ E	55°52′41″ N 23°28′08″ E	55°35′32″ N 24°02′58″ E
Collection date	13.vi.2015	12.vii.2015	12.vii.2015	03.vi.2017
Voucher number	EBPEM-7338	EBPEM-7340	EBPEM-7339	EBPEM-7546
DNA markers:				
*CO1–CO2–ATP8*	PP025829–35	MK625008	PP025835	MK625005
*ND6–CytB–ND1*	PP025836–40	MK757260	PP025841	MK757257
*18S*	PP059061	MK640428	PP059062	MK640425
*28S*	PP059063	MK640440	PP059064	MK640437
*ITS1*	PP059065	MK640432	PP059066	MK640429
*PB*	PP025842–43	MK628914	PP025844	MK628911
*AbdB*	PP025845	MK628918	PP025846	MK628915
*ArgK*	PP025847	MK628922	PP025848	MK628919
*Ube2g1*	PP025849	MK628926	PP025850	MK628923
*MRPP3*	PP025851	MK634482	PP025852	MK634479
*PTCD2*	PP025853	MK634478	PP025854	MK634475

**Table 3 insects-15-00086-t003:** Coefficients of discriminant (canonical) functions best segregating the morphometric measurements (see Section 2.2) of *Diodontus argillicola* sp. nov., *D. tristis* and *D. medius*. Scatterplots of canonical scores are in Figure 1.

Female				Male		
Measurement	Root 1	Root 2		Measurement	Root 1	Root 2
WH	−10.85	22.37		WH	−15.30	0.41
LID	39.82	11.30		LID	21.09	39.24
UID	6.06	19.43		UID	−24.19	−3.51
IMD	−36.18	−7.64		WCA	−20.32	11.00
WCA	25.94	4.19		POD	−27.62	8.22
WLA	−1.73	17.74		OOD	84.73	−45.76
LM	−3.02	−4.06		WHO	7.62	−62.84
POD	−55.25	−23.05		LF	3.19	16.15
OOD	10.52	−9.02		LSC	−16.19	−52.17
WHO	32.81	−68.44		3FL	25.11	2.19
LV	−14.31	−2.81		L6F	−20.06	38.94
LF	17.08	−35.31		W6F	1.93	−55.94
LCL	−33.64	31.40		PRN	7.61	−7.89
LSC	0.18	−8.46		COL	−5.92	5.09
3FL	−4.39	23.45		Constant	2.83	−1.22
PRN	14.13	−19.10				
COL	−5.61	6.87				
Constant	2.73	6.22				

## Data Availability

The data presented in this study are available in this article.

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
