# Peer review of "Ecological Speciation without Morphological Differentiation? A New Cryptic Species of Diodontus Curtis (Hymenoptera, Pemphredonidae) from the Centre of Europe†"

_insects, 2024, doi:10.3390/insects15020086_

Round 1

Reviewer 1 Report

Comments and Suggestions for Authors

This is an important paper, as it documents for the first time the presence of cryptic species within Sphecidae (sensu lato), recognizable mainly by molecular methods. The English of the text is quite good, and the rules of the Code of Zoological Nomenclature are observed. The authors know the pertinent literature very well.  The name of the new species is well chosen and euphonious.

I have the following critical comments:

1.      The title should be extended by the following: “, with a key to the Palearctic species of the Diodontus tristis species group”.

2.      The presence of the key should be mentioned in the Abstract.

3.      A definition of the D. tristis species group should be provided just before the key.

4.      In the key title, the word “Provisional” is unnecessary and should be deleted.

5.      The Etymology of the new species must be completed by: “and the Latin word cola, “inhabitant”, “dweller”. A noun in apposition to the generic name.”

6.      I suggest adding a section “Geographic Distribution”, including the text “Known from two localities in Lithuania”.

7.      Why is the new species known from only two localities? D. tristis was certainly collected in many other places in Lithuania. Was D. argyllicola not found in any other place? Or did the authors pay no attention to its possible presence elsewhere? In any case, its absence elsewhere deserves some discussion.

8.      How do you know that Vander Linden dealt with the species that you call D. tristis and not with D. argyllicola? You should provide an explanation for this point and demonstrate that your interpretation of this name is correct.

Author Response

1.      The title should be extended by the following: “, with a key to the Palearctic species of the Diodontus tristis species group”.

- The authors humbly consider the included identification key as a preliminary one, without yet illustrations of other little known species of D. tristis group (D. asiaticus, D. valkeilai), thus insufficiently novel and elaborated to be presented in the publication title. We also consider the earlier publication (Budrys et al., 2019) with similar content (two new species and their molecular phylogeny, with a preliminary key of another species group), where the identification key has not been mentioned in the title. Therefore, if the Editor and the reviewer would not insist, we would prefer to postpone the mentioning of identification key in the publication title until the further revision of Palearctic Diodontus, which is in preparation.

2.      The presence of the key should be mentioned in the Abstract.

- The key is mentioned in the Abstract (see the last sentence).

3.      A definition of the D. tristis species group should be provided just before the key.

- We added a paragraph with definition of the D.tristis group.

4.      In the key title, the word “Provisional” is unnecessary and should be deleted.

- Done.

5.      The Etymology of the new species must be completed by: “and the Latin word cola, “inhabitant”, “dweller”. A noun in apposition to the generic name.”

- Done. Thank you for important addition.

6.      I suggest adding a section “Geographic Distribution”, including the text “Known from two localities in Lithuania”.

- Done.

7.      Why is the new species known from only two localities? D. tristis was certainly collected in many other places in Lithuania. Was D. argyllicola not found in any other place? Or did the authors pay no attention to its possible presence elsewhere? In any case, its absence elsewhere deserves some discussion.

- We have added a paragraph of discussion on the distribution of the new cryptic species (see Discussion)

8.      How do you know that Vander Linden dealt with the species that you call D. tristis and not with D. argyllicola? You should provide an explanation for this point and demonstrate that your interpretation of this name is correct.

- Vander Linden's type series of Pemphredon tristis comes from Belgium (the exact locality is not mentioned in the description), however, the collection with types, to the knowledge of authors, is lost. The designation of a neotype, preferably from the type area, i.e. Belgium, is outside the scope of this study. The authors have added a paragraph of discussion considering the D. tristis types (see the Discussion).

Reviewer 2 Report

Comments and Suggestions for Authors

Dear authors,

I congratulate on this nice finding and preparation of the excellent manuscript. Overall the study design as well as manuscript preparation is excellently done and I could not find obvious mistakes to comment. Only small detail I can suggest is that there was hyphen missing at the beginning of the second statement of the second couplet (line 461) in the identification key.

Other than that I have just two concerns.

1) I understand that some of the COI barcodes used for reconstructing phylogeny illustrated at the figure 4 are and remain unpublished (mitotype ID labels including EBPEM). If that is so, then perhaps these sequences should still be published also.

2) Why do you assume, that the species inhabiting sandy habitats is already described by Vander Linden as Diodontus tristis and the species preferring clay walls is undescribed? It could just as easily be vice versa. I did not find any comments on studying type materials of D. tristis and moreover, with cryptic species it may not help at all. Alternatively there might be data about habitat, where type material of D. tristis was collected and you may rely on that.

Additionally I have two suggestions.

1) It would be useful to connect the specimens listed as type material (holotype and paratypes) with the sequences/mitotypes used for reconstructing phylogeny. It could be done by adding ID numbers to the list of type material. If mitotype numbering is not identical with specimen voucher numbers (eg more than one specimen correspond with one mitotype), then perhaps a table can be prepared. Maybe supplementary file where basic data is accumulated (occurrence data, depository and ID).

2) I had a look at the Estonian specimens in order to see if I can separate these two species (in case both occur in Estonia of course). Without DNA I can't say anything conclusive but most of Estonian specimens originate from vertical substrates (clay walls or sandstone cliffs) and only few (just one female) from sandy dunes. Although I had only one female from sandy habitat I noticed it had somewhat more slender antennas than all the other specimens. I did not notice that you had measured antennal segments in the morphometric analyses but my suggestion is if you already haven't tried this perhaps measuring proportions (length/with) of central antennal segment (for example the fifth segment) would be useful. This may not be productive at all but you can have a look at specimens separated with the help of DNA barcoding and decide if it deserves the effort.

Sincerely,

Author Response

1) I understand that some of the COI barcodes used for reconstructing phylogeny illustrated at the figure 4 are and remain unpublished (mitotype ID labels including EBPEM). If that is so, then perhaps these sequences should still be published also.

- The authors have included the voucher specimen numbers in the Type material list. The corresponding mitotype sequences of the voucher specimens have been uploaded to the GenBank, their accession numbers are presented in the Table 2.

2) Why do you assume, that the species inhabiting sandy habitats is already described by Vander Linden as Diodontus tristis and the species preferring clay walls is undescribed? It could just as easily be vice versa. I did not find any comments on studying type materials of D. tristis and moreover, with cryptic species it may not help at all. Alternatively there might be data about habitat, where type material of D. tristis was collected and you may rely on that.

- Right, we cannot be sure, in particular having in mind that the type series of Pemphredon tristis Vander Linden, 1829 is lost together with Vander Linden's collection, and the exact type locality is not published in the description. The authors suppose that the neotype of D. tristis should be designated from the fresh material collected in the type country (Belgium), with preliminary study of the molecular characters, at least the CO1 barcode. The authors have added a paragraph of discussion on this issue (see Discussion).

Additionally I have two suggestions.

1) It would be useful to connect the specimens listed as type material (holotype and paratypes) with the sequences/mitotypes used for reconstructing phylogeny. It could be done by adding ID numbers to the list of type material. If mitotype numbering is not identical with specimen voucher numbers (eg more than one specimen correspond with one mitotype), then perhaps a table can be prepared. Maybe supplementary file where basic data is accumulated (occurrence data, depository and ID).

- We added the ID numbers to the list of the type material. The mitotype sequences used in the study are uploaded in the GenBank; their accession numbers are presented in the Table 2. The CO1 mitotype numbering in Figure 4 differs from the voucher numbering, because it includes the DNA extraction and PCR product numbers (there were several PCR products from one voucher specimen). These PCR product samples (plastic tubes with amplified DNA) with their unique numbers are separately preserved in the NRCV collection. In the Figure 5a, the combined sequences (including several PCR products) were analysed, therefore, there are solely the voucher specimen ID numbers included. The publication of occurrence data of all studied Diodontus tristis, D. medius and other species is outside the scope of this study, these data will be included in a further revision of Diodontus, which is in preparation. 

2) I had a look at the Estonian specimens in order to see if I can separate these two species (in case both occur in Estonia of course). Without DNA I can't say anything conclusive but most of Estonian specimens originate from vertical substrates (clay walls or sandstone cliffs) and only few (just one female) from sandy dunes. Although I had only one female from sandy habitat I noticed it had somewhat more slender antennas than all the other specimens. I did not notice that you had measured antennal segments in the morphometric analyses but my suggestion is if you already haven't tried this perhaps measuring proportions (length/with) of central antennal segment (for example the fifth segment) would be useful. This may not be productive at all but you can have a look at specimens separated with the help of DNA barcoding and decide if it deserves the effort.

- We have measured the antennal segments (the measures LSC, 3FL, L6F, and W6F) and found very small, though statistically significant differences (slightly longer flagellum in both sexes of D. tristis, comparing to that of D. argillicola sp.n.), which are presented in the paragraph "Morphometric differences of Diodontus argillicola sp. nov. from D. tristis", just before the Description (page 13 of the final manuscript version).

Reviewer 3 Report

Comments and Suggestions for Authors

This study elaborates on one further case of probable cryptic speciation, in a small but widespread group of apoid wasps. The new, previously unrecognized species seems to be distinctive for its specific choice of nesting substrate (as compared to its closest relative). However, its separate specific status and phylogenetic position were possible to corroborate only by means of molecular evidence, while classical morphology and morphometric algorithms could not provide decisive separation.

While there is an ongoing debate regarding the merits and various problems related to "minimalist taxonomy approach", particularly on a large scale (i.e. in dealing with global taxonomic impediment), the study like this is undoubtedly justified. Authors applied the relevant methodology and conducted a number of analyses which seems to corroborate the hypothesis of existence of the separate cryptic species; they also provided discussion of alternative explanations and possible methodological problems inherent to this kind of approach.

Without a possibility to evaluate all the methodological details and numerous resulting parameters (which is partly beyond my field of expertise), I find that the study is relevant, properly conducted and scientifically sound – within the scope defined. Therefore, I recommend this MS to be published. I also suggest several minor to moderate corrections, which are more concentrated in some parts of the text than the others; some of these indicate that the level of English language expertise was not evenly performed while writing different sections.

Specific comments and suggestions:

Line 4: "the centre of Europe" sounds attractive for the title, but maybe somewhat inappropriate – as Lithuania is quite marginal to the central Europe; maybe to consider some more accurate geographic positioning?

Line 10: better omit "in nature".

Line 10-11: "The difficult to recognize species pairs or groups, called cryptic species, ..." may be better phrased; consider: "Species pairs or groups which are difficult to recognize, known as cryptic species, ...".

Line 12: consider rephrasing "ecosystem functions" with "functional role in ecosystems"; also, better omit "Therefore" and start the sentence with "It is".

Line 13: rephrase "diversity" with "other diversity metrics".

Line 16-17: the statement "; however, it presumably has a different, specific nesting habitat" better fits at the end of the next sentence, as "; presumably, it differs also by nesting habitat."

Line 19: "of Lithuania" should be "in Lithuania".

Line 20: rephrase "typical mitotypes of the species" as "the specimens with typical mitotypes"

Line 27: instead of "very little" should be "only subtle".

Line 30: instead of "secure identification, the phylogenetic position of it inside the genus,", it is better "confident identification, its phylogenetic position within the genus,".

Line 32: change "habitat differentiation" to "differentiation by habitat".

Line 48: "Curtis, 1834" should not be in parentheses.

Line 52-54: the wording of the following text is not fully correct: "fulfil ecosystem services as important natural enemies of herbivorous insects and, to a lesser extent, as pollinators" should be rephrased in several points: "provide ecosystem services of biological control (as important natural enemies of herbivorous insects) and, to a lesser extent, of pollination".

Line 60: "D. insidiosus Spooner, 1938 complex" – when species complexes or other informal taxonomic groupings are mentioned in such a context, it is not necessary (while also sounds cumbersome) to cite the authorship/year for the nominal species – at least year should be omitted.

Line 65: "similarity" should be specified, e.g. "ecological similarity".

Lines 82-86: the whole paragraph "The aims..." is not appropriately organized and connected to the preceding text of the Introduction! After the general considerations of the topics of cryptic species, their possible origins and various methodological issues of their recognition etc., there should be some short explanation of how have authors come about the discovery of this new case, leading to the study (and its aims). In the proper "storyline" the applied methodology/approach should precede the results (=species description and phylogeny).

Line 83: "from the" should better change into: "in the" or "of the".

Line 83: "Diodontus tristis (Vander Linden, 1829) species group" – the same comment regarding author/year as for the line 60.

Lines 89-93: any mentioning of "type specimens" in the context of "Materials" is inappropriate – you were collecting the specimens for the study, which happened to become the type-series only upon thorough examination and formal description (i.e., not before the section 3.4.); accordingly, the whole sentence about holotype deposition also better fits in that section!

Lines 246-247: "sister species of" should be "sister species to".

Lines 249: replace "distinctness" with "distinctiveness".

Lines 339-342: the part of this long sentence should be improved, starting with exclusion of "type series" term from considerations of specimens before they became types (=this happened only after examination and the description!), while "may also" is unnecessary weak statement; therefore, it could read as "The two species seem to differ also behaviorally, regarding their choice of nesting substrate: most of the specimens of the new species has been collected from the colony nesting in a clay wall, while all barcoded specimens from the colonies nesting in sand or gravel substrate happened to be D. tristis.".

Line 461: it is better "Central", instead of "Mid".

Lines 472-473: "in the middle part" should better be "in the middle" or "in the median part", or simply "medially".

Lines 474-475: correct "spiracular lobe black; if they with ivory spots", as follows: "spiracular lobe mostly black; if with ivory spots".

Line 522: the part "...speciation and species ecological adaptation and/or..." should be improved: "...speciation, ecological adaptation, and/or...".

Lines 521-525: the sentence spanning these lines is by far too long; should be cut into two, e.g. after "...differentiation" (line 523); instead of "for instance, ...", the better start of new sentence is "For example,".

Line 528: consider improvement of "gaps in it with application of species...", by rephrasing into: "gaps, in combination with species...".

Lines 568-570: this sentence needs improvement, e.g. "Based on assessments of anthropogenic pressures on digger wasp communities in Central Europe, Diodontus tristis has been recorded as common, or even dominant species of early-successional habitats.".

Comments on the Quality of English Language

(mentioned briefly in the previous section of comments)

Author Response

Line 4: "the centre of Europe" sounds attractive for the title, but maybe somewhat inappropriate – as Lithuania is quite marginal to the central Europe; maybe to consider some more accurate geographic positioning?

- Actually, Lithuania is at the midpoint of the European continent, estimated using one of methods, the centre of gravity, by the French National Geographic Institute (https://en.wikipedia.org/wiki/Geographical_midpoint_of_Europe#Lithuania). That is the method, when you find the equilibrium point of Europe that is cut out of the globe and put on a needle. Considering this fact, the authors prefer to keep these words in the title, trying to keep attractivity to the readership.

Line 10: better omit "in nature".

- Corrected.

Line 10-11: "The difficult to recognize species pairs or groups, called cryptic species, ..." may be better phrased; consider: "Species pairs or groups which are difficult to recognize, known as cryptic species, ...".

- Corrected, thank you.

Line 12: consider rephrasing "ecosystem functions" with "functional role in ecosystems"; also, better omit "Therefore" and start the sentence with "It is".

- Corrected.

Line 13: rephrase "diversity" with "other diversity metrics".

- Corrected.

Line 16-17: the statement "; however, it presumably has a different, specific nesting habitat" better fits at the end of the next sentence, as "; presumably, it differs also by nesting habitat."

- Corrected.

Line 19: "of Lithuania" should be "in Lithuania".

- Corrected.

Line 20: rephrase "typical mitotypes of the species" as "the specimens with typical mitotypes"

- Corrected.

Line 27: instead of "very little" should be "only subtle".

- Corrected.

Line 30: instead of "secure identification, the phylogenetic position of it inside the genus,", it is better "confident identification, its phylogenetic position within the genus,".

- Corrected.

Line 32: change "habitat differentiation" to "differentiation by habitat".

- Corrected.

Line 48: "Curtis, 1834" should not be in parentheses.

- Corrected (sorry for the mistake).

Line 52-54: the wording of the following text is not fully correct: "fulfil ecosystem services as important natural enemies of herbivorous insects and, to a lesser extent, as pollinators" should be rephrased in several points: "provide ecosystem services of biological control (as important natural enemies of herbivorous insects) and, to a lesser extent, of pollination".

- Corrected.

Line 60: "D. insidiosus Spooner, 1938 complex" – when species complexes or other informal taxonomic groupings are mentioned in such a context, it is not necessary (while also sounds cumbersome) to cite the authorship/year for the nominal species – at least year should be omitted.

- Rephrased to "complex of species related to D. insidiosus Spooner, 1938".

Line 65: "similarity" should be specified, e.g. "ecological similarity".

- Corrected to "general biological similarity", as it may include also the phylogenetic relatedness, similar morphological adaptations, ontogeny and life cycle, ecology, and behaviour.

Lines 82-86: the whole paragraph "The aims..." is not appropriately organized and connected to the preceding text of the Introduction! After the general considerations of the topics of cryptic species, their possible origins and various methodological issues of their recognition etc., there should be some short explanation of how have authors come about the discovery of this new case, leading to the study (and its aims). In the proper "storyline" the applied methodology/approach should precede the results (=species description and phylogeny).

- The paragraph considering the aims was shortened and rephrased, linking to the previous paragraph, which was also changed, trying to keep a proper "storyline" of the Introduction.

Line 83: "from the" should better change into: "in the" or "of the".

- Corrected.

Line 83: "Diodontus tristis (Vander Linden, 1829) species group" – the same comment regarding author/year as for the line 60.

- Corrected.

Lines 89-93: any mentioning of "type specimens" in the context of "Materials" is inappropriate – you were collecting the specimens for the study, which happened to become the type-series only upon thorough examination and formal description (i.e., not before the section 3.4.); accordingly, the whole sentence about holotype deposition also better fits in that section!

- Corrected: mentioning of "type specimens" and "holotype" removed.

Lines 246-247: "sister species of" should be "sister species to".

- Corrected.

Lines 249: replace "distinctness" with "distinctiveness".

- Corrected.

Lines 339-342: the part of this long sentence should be improved, starting with exclusion of "type series" term from considerations of specimens before they became types (=this happened only after examination and the description!), while "may also" is unnecessary weak statement; therefore, it could read as "The two species seem to differ also behaviorally, regarding their choice of nesting substrate: most of the specimens of the new species has been collected from the colony nesting in a clay wall, while all barcoded specimens from the colonies nesting in sand or gravel substrate happened to be D. tristis.".

- Corrected according with the suggestion - thank you.

Line 461: it is better "Central", instead of "Mid".

- Corrected.

Lines 472-473: "in the middle part" should better be "in the middle" or "in the median part", or simply "medially".

- Corrected.

Lines 474-475: correct "spiracular lobe black; if they with ivory spots", as follows: "spiracular lobe mostly black; if with ivory spots".

- "mostly black" would be not true, as the commonest D.tristis usually has ivory spots. Corrected explicitly listing the species with each of character state.

Line 522: the part "...speciation and species ecological adaptation and/or..." should be improved: "...speciation, ecological adaptation, and/or...".

- Corrected.

Lines 521-525: the sentence spanning these lines is by far too long; should be cut into two, e.g. after "...differentiation" (line 523); instead of "for instance, ...", the better start of new sentence is "For example,".

- The sentence was rephrased, trying to make it more clear. It could not be split into two, because it includes a single statement.

Line 528: consider improvement of "gaps in it with application of species...", by rephrasing into: "gaps, in combination with species...".

- Corrected.

Lines 568-570: this sentence needs improvement, e.g. "Based on assessments of anthropogenic pressures on digger wasp communities in Central Europe, Diodontus tristis has been recorded as common, or even dominant species of early-successional habitats.".

- Rephrased, trying to make the paragraph more clear.